# α1AMP-Activated Protein Kinase Protects against Lipopolysaccharide-Induced Endothelial Barrier Disruption via Junctional Reinforcement and Activation of the p38 MAPK/HSP27 Pathway

**DOI:** 10.3390/ijms21155581

**Published:** 2020-08-04

**Authors:** Marine Angé, Diego Castanares-Zapatero, Julien De Poortere, Cécile Dufeys, Guillaume E. Courtoy, Caroline Bouzin, Rozenn Quarck, Luc Bertrand, Christophe Beauloye, Sandrine Horman

**Affiliations:** 1Pôle de Recherche Cardiovasculaire (CARD), Institut de Recherche Expérimentale et Clinique (IREC), Université Catholique de Louvain (UCLouvain), 1200 Brussels, Belgium; marine.ange@uclouvain.be (M.A.); diego.castanares@uclouvain.be (D.C.-Z.); julien.depoortere@uclouvain.be (J.D.P.); cecile.dufeys@uclouvain.be (C.D.); luc.bertrand@uclouvain.be (L.B.); christophe.beauloye@uclouvain.be (C.B.); 2Division of Intensive Care, Cliniques Universitaires Saint Luc, 1200 Brussels, Belgium; 3IREC Imaging Platform, Institut de Recherche Expérimentale et Clinique (IREC), Université Catholique de Louvain (UCLouvain), 1200 Brussels, Belgium; guillaume.courtoy@uclouvain.be (G.E.C.); caroline.bouzin@uclouvain.be (C.B.); 4Department of Chronic Diseases & Metabolism (CHROMETA), Laboratory of Respiratory Diseases & Thoracic Surgery (BREATHE), KU Leuven, 3000 Leuven, Belgium; rozenn.quarck@kuleuven.be; 5Division of Cardiology, Cliniques Universitaires Saint-Luc, 1200 Brussels, Belgium

**Keywords:** α1AMPK, sepsis, lipopolysaccharide, endothelial permeability, VE-cadherin, connexin 43, zonula occludens-1, heat shock protein 27, actin cytoskeleton

## Abstract

Vascular hyperpermeability is a determinant factor in the pathophysiology of sepsis. While, AMP-activated protein kinase (AMPK) is known to play a role in maintaining endothelial barrier function in this condition. Therefore, we investigated the underlying molecular mechanisms of this protective effect. α1AMPK expression and/or activity was modulated in human dermal microvascular endothelial cells using either α1AMPK-targeting small interfering RNA or the direct pharmacological AMPK activator 991, prior to lipopolysaccharide (LPS) treatment. Western blotting was used to analyze the expression and/or phosphorylation of proteins that compose cellular junctions (zonula occludens-1 (ZO-1), vascular endothelial cadherin (VE-Cad), connexin 43 (Cx43)) or that regulate actin cytoskeleton (p38 MAPK; heat shock protein 27 (HSP27)). Functional endothelial permeability was assessed by in vitro Transwell assays, and quantification of cellular junctions in the plasma membrane was assessed by immunofluorescence. Actin cytoskeleton remodeling was evaluated through actin fluorescent staining. We consequently demonstrate that α1AMPK deficiency is associated with reduced expression of CX43, ZO-1, and VE-Cad, and that the drastic loss of CX43 is likely responsible for the subsequent decreased expression and localization of ZO-1 and VE-Cad in the plasma membrane. Moreover, α1AMPK activation by 991 protects against LPS-induced endothelial barrier disruption by reinforcing cortical actin cytoskeleton. This is due to a mechanism that involves the phosphorylation of p38 MAPK and HSP27, which is nonetheless independent of the small GTPase Rac1. This results in a drastic decrease of LPS-induced hyperpermeability. We conclude that α1AMPK activators that are suitable for clinical use may provide a specific therapeutic intervention that limits sepsis-induced vascular leakage.

## 1. Introduction

Sepsis, a major global health problem [1,2,3], is defined as a dysregulated host response to an infection [4]. Its evolution toward multi-organ failure (MOF), which is known as a crucial predictor of survival [5], directly results from microcirculatory dysfunction [6,7,8]. The latter is highly dependent on the regulation of vascular extravasation [9], and the maintenance of intravascular volume remains one of the most important challenges for the clinical support of septic patients.

Dysregulation of transendothelial paracellular permeability is the main determinant of sepsis-induced vascular extravasation. This results from the disruption of inter-endothelial junctions (IEJs) and the disorganization of actin cytoskeleton [10]. IEJs include adherens junctions (AJs) and tight junctions (TJs), which are both composed of multiprotein complexes. Some of their components, such as vascular endothelial cadherin (VE-Cad) and zonula occludens-1 (ZO-1), form part of the IEJ structure per se, and thus, constitute critical modulators of paracellular permeability [10]. Specifically, VE-Cad is recognized as the most important gatekeeper of the endothelial barrier [11,12,13,14]. Other constituents play major roles in the organization of protein–protein complexes in plasma membranes. They include connexin 43 (Cx43), which can directly and indirectly affect TJ and AJ assembly [15,16,17]. Actually, Cx43 signaling microdomains can influence a wide range of junctional- and cytoskeleton-associated proteins [18,19] through mechanisms that involve direct protein–protein interactions and transcriptional regulation [19,20,21]. The complexes of IEJs are anchored to the actin cytoskeleton, which contribute to define the IEJs’ architecture [22,23]. Actin filaments can exert antagonist effects on IEJs integrity, depending on their polymerization activity, localization, and contractility. These actin bundles can be categorized into two main types: cortical actin fibers that support the IEJs’ anchorage to the plasma membrane, which are predominant at the basal state, and contractile transcellular filaments called stress fibers that promote IEJs disassembly and internalization by exerting pulling forces. Stress fibers are mostly formed in response to pro-inflammatory agents. Their contraction contributes to the onset of intercellular-gap formation [22,23,24]. The key mediators involved in actin-dynamic regulation, include the small heat shock protein 27 kDa (HSP27), which plays a critical a role in the regulation of actin polymerization [25,26]. Unphosphorylated HSP27 behaves like a filamentous actin (F-actin) cap-binding protein by inhibiting actin polymerization. Phosphorylation of several distinct serine residues (Ser15, Ser78, and Ser82) relieves its capping activity and promotes actin microfilament formation. HSP27 is phosphorylated by mitogen-activated protein kinase-activated protein kinases (MAPKAPK) 2 and 3, which are both direct substrates of p38 MAPK α and β [27]. The role of HSP27 phosphorylation in the regulation of endothelial permeability remains controversial [25,26,27,28,29]. In addition, small Rho GTPases are crucial orchestrators of actin organization and contractility [23,30]. RhoA promotes stress fibers contraction, while Rac1 reinforces IEJs’ anchorage to the plasma membrane by promoting the development of a cortical actin network [31,32]. Altogether, junctional and cytoskeletal proteins regulate paracellular flow through tight control of the opening-closure sequences of the intercellular space. In pro-inflammatory conditions, paracellular permeability is enhanced because macromolecules and leucocytes are recruited within underlying tissues to regulate pathogen invasion. Extreme sepsis conditions drive homeostasis towards endothelial dysfunction, formation of intercellular gaps, subsequent massive plasma leakage and impaired oxygen diffusion into peripheral tissues, causing MOF. Targeting mediators that contribute to the vascular barrier integrity might therefore improve support for septic patients.

AMP-activated protein kinase (AMPK) is a key metabolic master that has broad effects on cellular functions, including organization of IEJs [33,34,35] and actin cytoskeleton [36,37,38,39,40,41]. Its main endothelial isoform, α1AMPK, has been reported to regulate paracellular permeability [42,43,44,45,46] and, more specifically, sepsis-induced vascular leakage [45,46,47,48,49]. Our group previously demonstrated that the mortality of mice exposed to sublethal doses of lipopolysaccharide (LPS) was dramatically increased by α1AMPK invalidation. This was also shown to be associated with enhanced vascular permeability [47]. Moreover, we showed that AMPK activation by the AMP-mimetic 5-aminoimidazole-4-carboxamide ribonucleotide (AICAr) in wild-type mice can attenuate LPS-induced tissue edema formation. Other direct [33,47,48,49] and indirect [45,50,51] AMPK activators have already demonstrated their capacity to protect the vascular barrier in in vivo models of sepsis, and some are even associated with anti-inflammatory effects [49,50,51] and improved survival [49]. Among these activators, only metformin is currently used in clinical practice. However, its administration during sepsis remains controversial because of the potential evolution toward lactic acidosis [52]. By considering the urgency to improve medical support for septic patients [53], special attention should be paid to identifying new molecule targeted therapies that are clinically applicable to the particular context of sepsis.

In this study, we sought to further elucidate mechanisms through which AMPK regulates junctional assembly and permeability in LPS treated microvascular endothelial cells. Specifically, we examined the effect of the specific and direct pan-AMPK activator 991 on IEJs and cytoskeleton organization. We report for the first time that AMPK plays a critical role in the maintenance and assembly of IEJs by regulating Cx43 expression, HSP27 phosphorylation and subsequent actin cytoskeleton remodeling.

## 2. Results

### 2.1. α1. AMPK Preserves IEJs Integrity

#### 2.1.1. α1. AMPK Deficiency is Associated with Downregulation of VE-Cad, ZO-1, and Cx43 Expression

We first investigated whether α1AMPK regulates VE-Cad, ZO-1, and Cx43 expression in unstimulated and LPS-treated HMECs. Cells transfected with AMPKα1-targeting or scrambled control small interfering RNAs (siRNAs) were either treated or not treated with the best-available AMPK activator 991 (1 µM, 1 h), prior to LPS stimulation (50 µM, 6 h) (Figure 1a). While, AICAr can have off-targets, the 991 compound has been recently described and is a direct and more specific AMPK activator [54]. Its effect is mainly allosteric, inducing a change in AMPK conformation and subsequent stimulation. Unlike LPS, treatment by 991 activates AMPK in HMECs, as reflected by AMPK phosphorylation on Thr172 and acetyl-CoA carboxylase (ACC) phosphorylation on Ser79. α2AMPK is not expressed in HMECs. Transfection of α1AMPK-targeting siRNA abrogates α1AMPK expression and 991-mediated ACC phosphorylation (Figure 1b,c). Western blot analysis also reveals that VE-Cad, ZO-1, and Cx43 protein expression are significantly downregulated in α1AMPK-depleted cells, although they are not affected by 991 or LPS treatment (Figure 1b,c). Considering the importance of Cx43 in the formation and scaffolding of IEJs between endothelial cells, we subsequently utilized a specific siRNA-targeting Cx43 to characterize the impact of its deletion on VE-Cad and ZO-1 expression. The protein level of Cx43 is significantly lower in cells transfected with the siRNA anti-Cx43, compared to scramble-transfected cells (Figure 2a). Cx43 deficiency is associated with significantly reduced expression of VE-Cad and ZO-1 (Figure 2b,c). Accordingly, fluorescence analysis confirms the drastic loss of VE-Cad and ZO-1 signals at the edges of Cx43-silenced cells (Figure 2d,e). These results suggest that α1AMPK plays a critical role in the regulation of Cx43 expression and subsequent maintenance and assembly of TJs and AJs, independently of LPS treatment.

#### 2.1.2. AMPK Activation Prevents LPS-Induced Disruption of IEJs and Endothelial Barrier Dysfunction

LPS drastically impairs the integrity of VE-Cad (Figure 3a,b), ZO-1 (Figure 3c,d), and Cx43 (Figure 3e,f). Indeed, the linear shape of cell–cell junctions, which is observed in basal conditions, appears jagged and disconnected after LPS treatment. Accordingly, membrane staining quantification for VE-Cad, ZO-1, and Cx43 is significantly lower in LPS-treated cells, compared to control cells (Figure 3b,d,f). These changes are associated with the formation of intercellular gaps, which indicates endothelial barrier disruption. As we show that LPS does not affect VE-Cad, ZO-1, and Cx43 total protein expression (Figure 1), the decreased membrane staining of IEJs is likely due to their mechanical disruption and internalization. Next, we determined the effects of the small-molecule AMPK activator 991. The 991 compound does not affect IEJs organization in unstimulated HMECs. However, it reinforces Cx43, VE-Cad, and ZO-1 anchorage to the plasma membrane upon LPS stimulation and also prevents the appearance of intercellular gaps. This protective effect is lost in α1AMPK-depleted cells, indicating an AMPK-dependent action (Figure 3a–f). Notably, the absence of α1AMPK induces a drastic decrease of VE-Cad, ZO-1, and Cx43 signals at the periphery of cells, regardless of the conditions, which confirms our previous Western blot data (Figure 1b,c). Finally, the involvement of AMPK pathway in endothelial barrier integrity was demonstrated by measuring the clearance of HRP-coupled streptavidin through the HMEC monolayer. As expected, 991 treatment significantly reduces LPS-induced hyperpermeability (Figure 3g). To inactivate AMPK, we used the pan-AMPK inhibitor SBI0206965. AMPK inactivation significantly impairs endothelial barrier function and completely abrogates 991-mediated protection (Figure 3g).

### 2.2. α1. AMPK Induces Actin Cytoskeleton Remodeling

#### 2.2.1. AMPK Activation Modulates Actin Organization and Polymerization

Given that junctional complexes are proposed to associate with the actin cytoskeleton, considered as a key component of cell permeability, we then investigated the impact of AMPK activation on LPS-induced F-actin morphological changes. In resting HMECs, F-actin is mainly organized in peripherical beams within the cytoplasm (Figure 4a). After LPS stimulation, peripherally located F-actin is substituted by centralized and irregular stress fibers running across the cytoplasm (Figure 4a,b). This effect is even more pronounced in the absence of α1AMPK, despite a significant decrease in the fluorescent signal that corresponds to F-actin (Figure 4c). AMPK activation by 991 strongly reinforces peripheral actin bands, both in unstimulated cells and in cells treated with LPS. This effect is reduced in the absence of α1AMPK (Figure 4a–c).

#### 2.2.2. α1. AMPK Regulates Actin Polymerization via Activation of the p38 MAPK/HSP27 Pathway

Due to its pivotal role in actin polymerization, we investigated whether HSP27 was involved in the protective action of α1AMPK against LPS-induced endothelial barrier impairment. The total expression and phosphorylation sate of HSP27(Ser82) and p38 MAPK(Thr180/Tyr182) were measured in the presence or absence of 991, before LPS stimulation. The results show that 991 and LPS increase p38 MAPK phosphorylation, and can individually and additionally enhance HSP27 phosphorylation (Figure 5a,b). Unlike LPS, the effect of 991 on HSP27 phosphorylation is attenuated in α1AMPK-depleted cells (Figure 5a,b). It clearly depends on p38 MAPK activation since preincubation with the selective p38 MAPK inhibitor SB203580 significantly reduces signal intensity, regardless of the conditions (Figure 5c,d). Surprisingly, treatment with SB203580 increases MAPK p38 phosphorylation, possibly because of feedback activation induced by the inhibition of p38 MAPK activity (Figure 5c) [55]. Next, we examined the effect of direct inhibition of p38 MAPK on the actin cytoskeleton. SB203580 globally reduces F-actin content and impairs the well-organized peripheral actin network of unstimulated cells, which acts in favor of the formation of stress fibers across the cytoplasm (Figure 6a–c). More interestingly, p38 MAPK inactivation prevents 991-induced enrichment of thick actin bundles around the cell periphery of LPS-treated cells (Figure 6a–c). Altogether, these results indicate that α1AMPK activation modifies the cytoskeletal response to LPS injury by potentiating the p38 MAPK/HSP27 pathway. Accordingly, while basal permeability is barely affected by p38 MAPK inhibition, SB203580 completely prevents the protective effect of 991 (Figure 6d). Since p38 MAPK has been reported as a downstream target of Rac1, a small GTPase intimately involved in regulating cortical actin structures and endothelial barrier function [56], Rac1 activation was measured in the presence or absence of 991, before LPS stimulation. G-LISA analysis shows that LPS and 991 can individually enhance Rac1 activity, while there is no additional effect of the combined treatments (Figure 7a). Pre-incubation with NSC23766 inhibits Rac1 activity (Figure 7a) but does not significantly change 991—or LPS-induced HSP27 phosphorylation (Figure 7b), indicating that the activation of the p38 MAPK/HSP27 axis is independent of Rac1 in both conditions.

## 3. Discussion

In this study, we first demonstrate that α1AMPK is essential in maintaining the proper expression and architecture of IEJs in basal conditions. Second, we identify the p38 MAPK/HSP27 pathway and actin cytoskeleton as central mediators of the protective effect of activated α1AMPK on the endothelial barrier. Experiments were performed using HMECs, which constitute a highly relevant model for the assessment of vascular leakage mechanisms that mainly occur at the level of capillaries and postcapillary venules. The α1AMPK isoform plays a predominant role in these cells since its specific invalidation using a siRNA strategy almost completely abrogates the 991 effect on ACC phosphorylation, which is the bona fide substrate of AMPK. Several studies, including ours, have reported the beneficial role of AMPK in endothelial barrier regulation during sepsis [45,46,47,48,49]. Here, we confirm this protective action in microvascular endothelial cells that are exposed to LPS and highlight the involvement of new molecular mechanisms.

Our previous work has reported on the protective role of basal α1AMPK in the maintenance of endothelial barrier function, through the preservation of ZO-1 integrity in TJs [47]. Here, we demonstrate that, in addition to ZO-1, VE-Cad is also distorted and downregulated in the absence of α1AMPK. Our data further support the emerging concept that Cx43 can contribute to the basal expression and organization of ZO-1 and VE-Cad. Cx43, which is a protein normally associated with gap junctional communication, has recognizable scaffolding functions that contribute to various biological processes [16,19]. The AMPK-Cx43 pathway has been highlighted in cardiomyocytes [57] and in cardiac fibroblasts [58] in which it plays a role in the fibrotic response commonly seen with a variety of cardiac pathologies [59]. The link between AMPK and Cx43 has never been demonstrated in the endothelium. We show for the first time that AMPK deficiency is associated with a drastic loss of Cx43 in endothelial cells, which leads to the subsequent disruption of tight and adherens junctions. The fact that the 991 compound has no impact on Cx43 expression supports the notion that non-catalytic functions of AMPK are rather involved in this regulation. These non-catalytic functions may include the scaffolding of protein complexes, the competition for protein interactions, allosteric effects on other enzymes, or subcellular targeting. Among others, ERK1 and ERK2 were shown to influence their substrates not only by phosphorylation, but also by direct protein-protein interactions, independently of kinase activity [60]. We believe that this type of regulation likely explains how AMPK may influence Cx43 expression.

While the direct interaction of Cx43 with ZO-1 has been extensively demonstrated [17,21,61], the mechanism is more speculative for VE-Cad. The interaction between Cx43 and VE-Cad likely involves other partners, such as N-cadherin (N-Cad), p120, and α- and ß-catenin, all of them being known to bind Cx43 and promote VE-Cad stability [11,20,33,62,63,64]. In addition to its scaffolding action, Cx43 is also described as a modulator of gene transcription, by the interaction of its carboxy tail with transcription factors driving its translocation to the nucleus and binding to gene promoters [19,65]. However, these potential mechanisms do not exclude Cx43-independent transcriptional regulation of ZO-1 and VE-Cad by α1AMPK. Indeed, AMPK can directly (via the phosphorylation of transcriptional regulators) or indirectly (via the induction of sirtuins) regulate gene transcription [66]. Post-transcriptional (changes in micro-RNAs expression) and post-translational (acetylation or nitrosylation) mechanisms might also be involved [67,68,69,70]. Future experimental work is necessary to delineate the molecular regulation of VE-Cad, ZO-1, and Cx43 by AMPK.

Our present work emphasizes the critical role of actin cytoskeleton in the α1AMPK dependent protection of IEJ integrity in LPS-treated endothelial cells. Our observations are supported by data showing that, in epithelial and endothelial cells, activated AMPK can reinforce the interaction between cadherin-catenin complexes and actin cytoskeleton via both regulation of N-Cad [33,63] and phosphorylation of G-alpha interacting vesicle associated protein (GIV) [71,72]. Other protective mechanisms have been highlighted. In Caco-2 cells submitted to a calcium switch [34,35,73,74], α1AMPK activation is associated with a change in the expression level of VE-Cad, ZO-1, or Cx43 [34,42], whereas our present data show that the expression of these proteins is not changed in 991-treated HMECs. In other cellular models, activated α1AMPK may affect the integrity of TJs and AJs by phosphorylating some of their components. For example, claudin-1 and -4 have been described as direct AMPK substrates in epithelial Eph4 cells and SV40 immortalized rat submandibular acinar cell lines, respectively [44,75]. Afadin is another AMPK target involved in reinforcing TJs in Madin-Darby Canine Kidney (MDCK) cells, via its binding to ZO-1 [76].

We and others have previously demonstrated that α1AMPK regulates the organization of actin cytoskeleton in different cell types [36,37,38,39,40,41]. HSP27 plays an important role in actin dynamics, through its phosphorylation downstream of p38 MAPK [25,26,27]. Very interestingly, a link between Cx43 and HSP27 was previously demonstrated in HeLa cells [77]. In this model, Cx43 regulates actin dynamics and migration by binding p21-activated protein kinase 1 (PAK1), which then activates the downstream p38 MAPK/HSP27 pathway. Our data show that HSP27 phosphorylation protects the endothelial barrier function. However, the role of HSP27 in the regulation of endothelial permeability remains controversial. For example, HSP27 phosphorylation protects against hypoxia- or burn serum-induced permeability of endothelial cells [25,26,27,28,29]. On the contrary, pertussis toxin-induced endothelial permeability is temporally linked to p38 MAPK activation and phosphorylation of HSP27 [27,78]; and LPS-induced endothelial barrier dysfunction correlates with HSP27 phosphorylation in vitro (our own data) and in vivo [28,79]. The reason for these discrepancies is not clear. It has been suggested that activation of the p38 MAPK/HSP27 pathway by LPS might be an adaptive response, thereby promoting centralized stress fibers and subsequent cell adhesion, in order to counteract contractility. This is rather regulated by the RhoA pathway and downstream myosin light chain phosphorylation [25].

Given that p38 MAPK is a downstream effector of AMPK in endothelial cells [80], we postulated that it could be involved in the protective action of 991 on cell cytoskeleton and permeability. Subsequently, we asserted that HSP27 phosphorylation was significantly increased in 991-treated HMECs. While, LPS is associated with the formation of centralized stress fibers, 991 increases cortical actin structures, suggesting the likely involvement of two different p38 MAPK activation pathways. Typically, p38 MAPK is activated through small GTPases, such as Rac1, and the downstream canonical three-tiered kinase cascade [81,82]. Rac1 is activated by LPS and 991 to the same extent, but surprisingly, we observed that its inhibition had no significant impact on LPS- or 991-induced HSP27 phosphorylation. This suggests the involvement of other non-canonical mechanisms of p38 MAPK activation. Accordingly, it was recently shown that the p38 isoform in HMECs can be autoactivated through its interaction with the transforming growth factor α-activated kinase 1 binding proteins (TAB) 1, 2, or 3, which is independent of the MAPK cascades [83]. Notably, the direct activation of p38 MAPK by AMPK, via the scaffold protein TAB1, has been previously described in apoptotic lymphocytes and the ischemic heart [84,85], strengthening the relevance of this mechanism in 991-treated HMECs. Finally, a different non-canonical pathway for p38 MAPK activation can be mediated through autophosphorylation, which is facilitated by the tyrosine kinase Zap70 [83,86,87]. The role of these non-canonical mechanisms of p38 MAPK, in the regulation of endothelial barrier disruption, remains poorly understood and warrants further exploration.

In summary, our data show that α1AMPK is required to maintain the integrity of TJs and AJs in microvascular endothelial cells. Moreover, α1AMPK activation leads to the phosphorylation of HSP27 via a non-canonical p38 MAPK dependent mechanism, which results in the reinforcement of the submembrane actin network during LPS challenge, contributing to the consolidation of IEJs in the plasma membrane and intercellular tethering (Figure 8). Therefore, α1AMPK activation might have therapeutic potential to alleviate conditions in which microvascular barrier formation is dramatically compromised, such as sepsis [88].

## 4. Materials and Methods

### 4.1. Reagents and Antibodies

The 991 compound was kindly provided by Benoit Viollet (Institut Cochin, DR2 Inserm, Paris, France). SBI0206965 (#SML1540), SB203580 (#S8307), NSC23766 (#SML0952), and LPS O55:B5 (50 μg/mL, #L2880) were from Sigma-Aldrich (Overijse, Belgium). We also used a halt protease and phosphatase inhibitor cocktail (#78442; Thermo Fisher Scientific, Waltham, MA, USA), Lipofectamine RNAiMAX reagent (#13778-150; Invitrogen, Thermo Fisher Scientific), Opti-MEM (#31985070; Gibco, Thermo Fisher Scientific), siRNA negative control (#AM4635; Ambion, Thermo Fisher Scientific), siRNA PRKAA1 (#AM51334; Ambion), siRNA GAJ1 (#4392422; Ambion), BM chemiluminescence blotting system (#11500694001; Roche, Sigma-Aldrich), 12 mm μ-dish culture inserts (634-1577P; VWR), Alexa Fluor 488 Deoxyribonuclease I (DNAse I) (#D12371; Invitrogen), Alexa Fluor 568 Phalloidin (#A12380; Invitrogen), Transwell inserts (#3413; Corning, Sigma-Aldrich), M200 medium (#M200500; Gibco), HRP-coupled streptavidin (#15:1000, DY998; R&D Systems, Minneapolis, MN, USA), TMB substrate (#T040; Sigma-Aldrich), H_2_SO_4_ (#30743; Sigma-Aldrich), and RAC1 G-LISA kit (#BK128; Cytoskeleton, Inc., Denver, CO, USA). The antibodies used were α1AMPK (#MA5-15815; Thermo Fisher Scientific), phospho-AMPK Thr172 (#2535; Cell Signaling Technology, Danvers, MA, USA), phospho-ACC S79 (#3661; Cell Signaling Technology), Cx43 (#3512; Cell Signaling Technology), VE-Cad (#36-1900; Thermo Fisher Scientific), ZO-1 (#339100; Thermo Fisher Scientific), p38 MAPK (#9212; Cell Signaling Technology), phospho-p38 MAPK T180/Y182 (#9211; Cell Signaling Technology), heat shock protein of 27 kDa (HSP-27) (#2402; Cell Signaling Technology), phospho-HSP27 S82 (#44534; Life Technologies, Thermo Fisher Scientific), eEF2 (#PA5-17794; Thermo Fisher Scientific), secondary horseradish peroxidase (HRP)-conjugated antibodies (#A0545; Sigma-Aldrich or#554002; BD Biosciences, San Jose, CA, USA), and Alexa Fluor-coupled secondary antibodies (#A21202 or #A21206; Invitrogen).

### 4.2. Cell Culture and Treatments

HMECs were purchased from Promocell (C-12225) and cultured according to the manufacturer’s recommendations, using Endothelial Cells Growth Medium MV (C-22020; Promocell, Heidelberg, Germany) containing 1% penicillin-streptomycin, at 37 °C and 5% CO_2_ in a humidified incubator. The cells were sub-cultured when reaching 80% confluence and used until subculture number 7. To proceed with the experiments, cells were seeded at a density of 10^4^/cm^2^ and cultured for 48–72 h, until total confluence was reached. Medium deprivation was performed for two hours prior to treatment or experimentation.

### 4.3. siRNA Transfections

For endothelial α1AMPK and Cx43 silencing, HMECs were seeded the day before transfection with heparin-free Endothelial Cells Growth Medium MV2 (C-22022; Promocell) to reach 60% confluence within 24 h. Reverse transfection was then performed for 48 h with either a control non-targeting siRNA construct (50 nM), a siRNA specifically targeting PRKAA1 (50 nM), or GAJ1 (50 nM). This was performed using a lipofectamine RNAimax transfecting reagent, which adhered to the manufacturer’s instructions.

### 4.4. Western Blotting

Western blot analysis was performed on HMEC lysis, as previously described [89]. Protein content was measured using the Bradford method, which used bovine serum albumin (BSA) as a reference. Proteins (15 μg) were separated by sodium dodecyl sulfate-polyacrylamide gel electrophoresis and electroblotted. Membranes were then probed with the primary antibody overnight at 4 °C, in order to assess protein phosphorylation or total expression level. Primary antibodies included α1AMPK (1:1000), phospho-AMPK Thr172 (1:1000), phospho-ACC (1:5000), Cx43 (1:1000), VE-Cad (4:1000), ZO-1 (2:1000), p38 MAPK (1:1000), phospho-p38 MAPK T180/Y182 (1:1000), HSP27 (1:1000), phospho-HSP27 S82 (1:1000), and eEF2 (1:1000). Membranes were then incubated with appropriate secondary HRP-conjugated antibodies (1:20,000, #A0545 or 1:10,000, #554002; both Sigma-Aldrich) for one hour at room temperature. A chemiluminescence blotting system was used for detection. Quantification was executed by ImageJ software and normalized with eEF2 as a loading control on the same gel. Each experiment was repeated three times.

### 4.5. Immunofluorescence Microscopy

HMECs were seeded on non-coated glass coverslips at 20 × 10^3^ cells/cm^2^, 72 h before treatment. After treatment, cells were fixed in 4% paraformaldehyde, permeabilized with 0.3% triton X-100 for ten minutes, and blocked with 10% BSA for 45 min. For intercellular junction studies, cells were stained as previously described [47], using the following primary antibodies: Cx43 (1:100), VE-Cad (1:25), or ZO-1 (1:50). Cells were then incubated with the appropriate Alexa Fluor-coupled secondary antibody (1:1000, #A21202 or #A21206). For cytoskeleton studies, HMECs were incubated with Alexa Fluor 568 Phalloidin (150 nM) for 15 min at room temperature. Nuclei were stained with 4′,6-diamidino-2-phenylindole (DAPI). Stainings were visualized under a Zeiss Imager Z1 microscope that was equipped with an ApoTome device. Pictures were acquired with a 20× objective. Each experiment was repeated three times.

### 4.6. Image Analysis

Quantitative image analysis was performed on uncompressed images (native format: zvi) with Fiji 1.52n on MacOS (10.14.5). The intercellular junctions evidenced by ZO-1 or VE-Cad staining were automatically delimited by a fixed-value threshold method. The stained area was quantified, and the mean signal intensity was calculated within this selection. Stained membrane segments were subsequently detected using the Analyze Particles and Skeletonize tools, and automatically counted. Due to the irregular distribution of the Cx43 signal, all cell membranes were manually defined as regions of interest. The Cx43 positive area and mean signal intensity were then quantified within these regions.

To analyze cytoskeleton organization, F-actin and nuclei were sequentially detected using threshold methods in the red and blue channels, respectively. Stained area and integrated density were measured for F-actin. The area fraction of F-actin that was colocalized within nuclei areas was subsequently calculated and defined as stress fibers. The mean signal intensity was also measured.

For normalization purposes, the nuclei in all images were automatically counted using a threshold method and the Analyze Particles tool.

### 4.7. Endothelial Permeability Assessment

For the endothelial permeability assay, HMECs (10^5^ cells/well) were seeded on gelatin-coated Transwell inserts of 24-well plates, in 250 μL complete with Endothelial Cells Growth Medium MV. They were then incubated for 72 h at 37 °C and with 5% CO_2_. The cells were incubated in free M200 medium for two hours before stimulation. The cells were then incubated with the different compounds, as indicated in the figure legends. After treatment, the medium from the upper chamber was replaced by 300 μL of M200, containing HRP-coupled streptavidin. The medium of the lower chamber was collected after ten minutes of incubation at 37 °C, and every condition was aliquoted in triplicate. The TMB substrate was added for ten minutes, and an Elisa reader was used to stop the reaction with 2N H_2_SO_4_ before acquiring 450 nm absorption. Resultant absorption intensity values were normalized over the vehicle control condition. Each experiment was repeated three times.

### 4.8. Small GTPases Activity Assay

Rac1 activity was measured in protein lysates from HMECs using the commercially available G-LISA assay kit, according to the manufacturer’s instructions. Each experiment was repeated three times.

### 4.9. Statistical Analysis

Statistical analyses were conducted using SPSS v.25 software (IBM Corp., Armonk, NY, USA) and graphs build with GraphPad Prism 7.0 (GraphPad Software, La Jolla, CA, USA). All tests were two-sided, with significance set at the 0.05 probability level. Data were expressed as mean ± standard deviation. Means were compared using an unpaired Student’s *t*-test. One- or two-way analysis of variance with *f* test was used in comparisons wtih more than two groups. The Bonferroni correction was used for multiple comparisons.

## Figures and Tables

**Figure 1 ijms-21-05581-f001:**
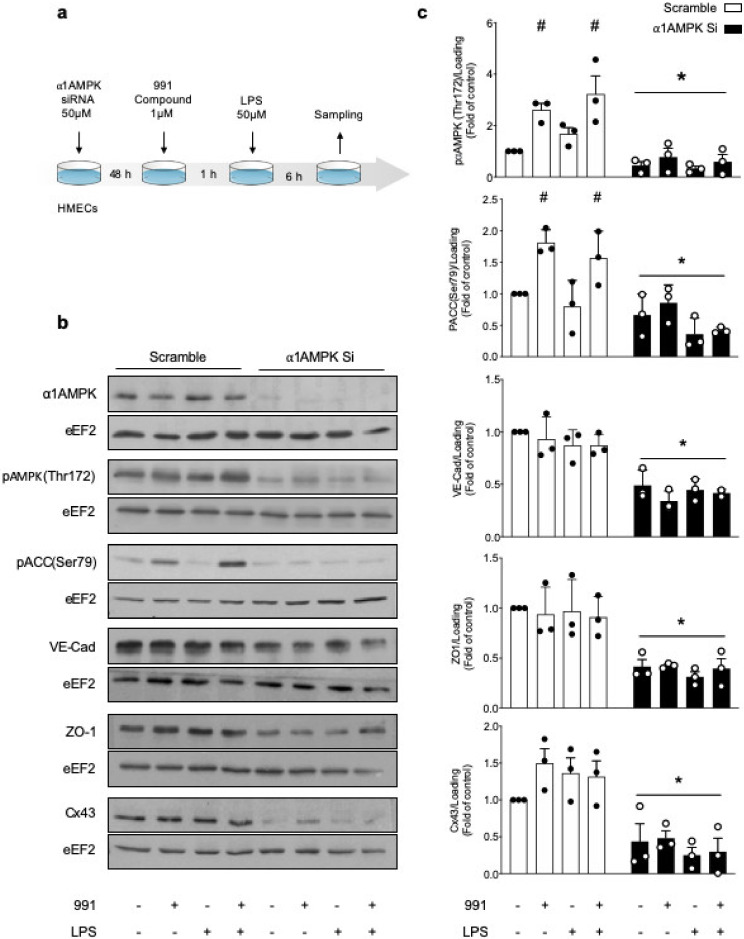
Basal α1AMPK regulates VE-Cad, ZO-1, and Cx43 expression. (**a**) Schematic representation of the experimental design, in which HMECs were transfected with control non-targeting siRNA or α1AMPK siRNA (50 nM) for 48 h. Then, they were treated with 991 (1 μM) or DMSO for one hour and subsequently exposed or not exposed to lipopolysaccharide (LPS, 50 μM) for six hours; (**b**,**c**) Human dermal microvascular endothelial cells (HMECs) were treated according to the protocol detailed in (**a**). Cell lysates were submitted to Western blot analysis and probed with α1AMPK, phospho-AMPK (Thr172), phospho-ACC (Ser79), VE-Cad, ZO-1, and Cx43 antibodies (Abs). Anti-eukaryotic elongation factor 2 (eEF2) was used as a loading control. Representative western blots; (**b**) and quantifications (**c**) are shown. Data are expressed as mean ± standard deviation (SD) (three biological replicates for each condition). ^#^
*p* < 0.05 is relative to corresponding untreated HMECs and * *p* < 0.05 is relative to cells transfected with the scrambled siRNA. The data underwent two-way analysis of variance (ANOVA).

**Figure 2 ijms-21-05581-f002:**
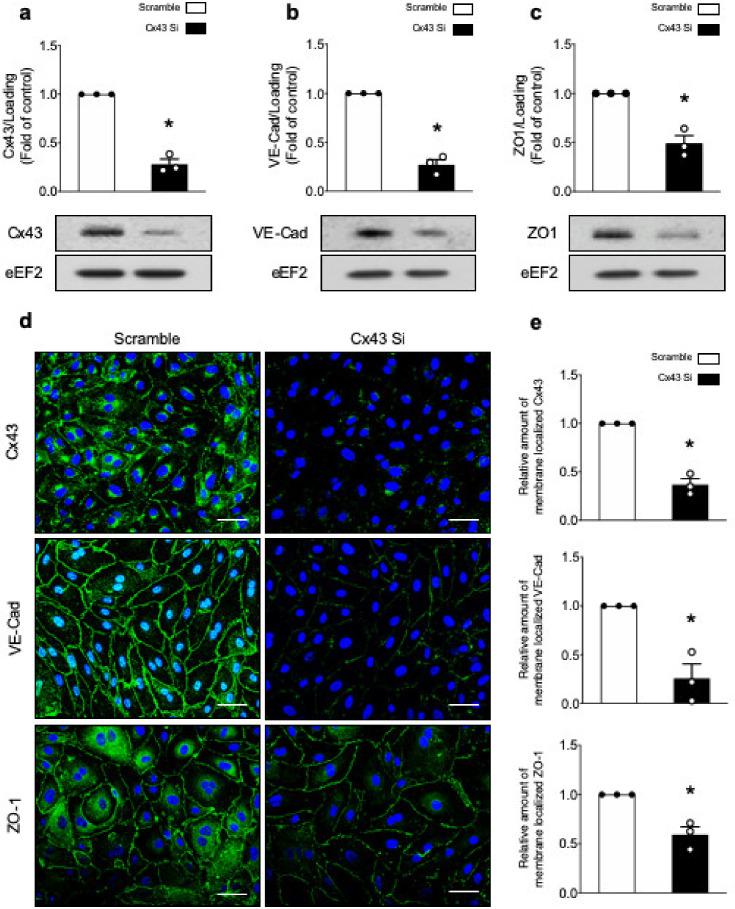
Cx43 deficiency is associated with decreased VE-Cad and ZO-1 expression. (**a**–**e**) HMECs were transfected with control non-targeting siRNA or Cx43 siRNA (50nM) for 48 h. (**a**–**c**) Cell lysates were submitted to Western blot analysis and probed with Cx43 (**a**), VE-Cad (**b**), and ZO-1 (**c**) Abs. eEF2 was used as a loading control. Representative Western blots (**top** panel) and quantifications (**bottom** panel) are shown; (**d**,**e**) Cx43, VE-Cad, and ZO-1 immunostainings. Nuclei are stained with DAPI. Scale bar, 50 μm. Representative images (**d**) and quantifications (**e**) are shown. Data are expressed as mean ± SD (three biological replicates for each condition). * *p* < 0.05 is relative to cells transfected with the scrambled siRNA. Analyses were performed using Student’s *t*-test.

**Figure 3 ijms-21-05581-f003:**
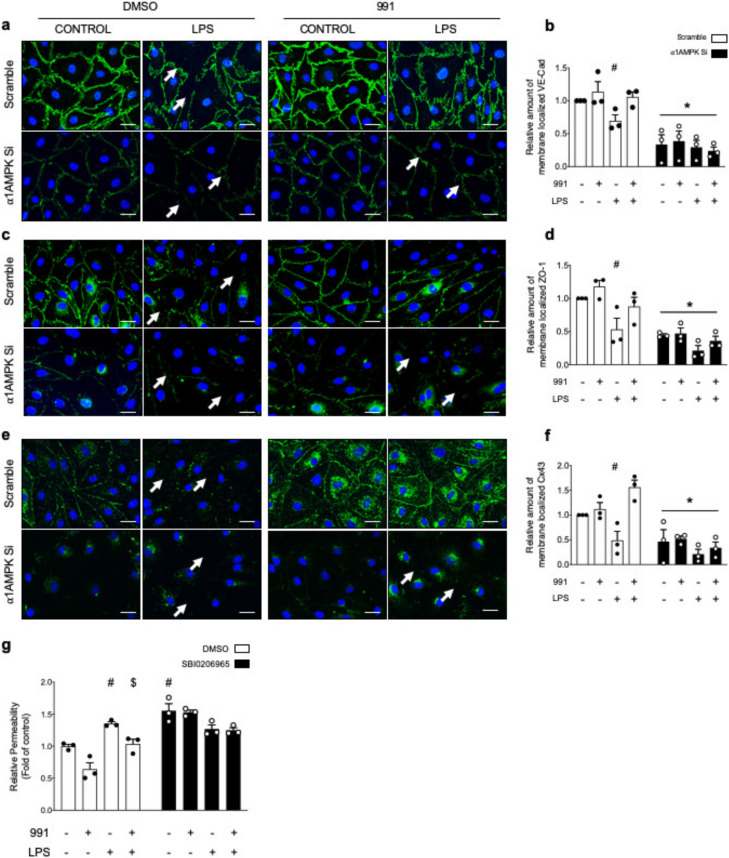
AMPK Activation Preserves IEJs Organization and Endothelial Barrier Function in HMECs Under LPS Challenge. (**a**–**f**) HMECs were transfected with control non-targeting siRNA or α1AMPK siRNA (50 nM) for 48 h. Then, HMECs were treated with DMSO or 991 (1 μM) for one hour, after which they were, either exposed or not exposed to LPS (50 μM) for six hours. Immunostainings and membrane-staining quantification of VE-Cad (**a**,**b**), ZO-1 (**c**,**d**), and Cx43 (**e**,**f**) are shown. Intercellular gaps are indicated by white arrows. Nuclei are stained with DAPI. Scale bar, 50 μm. Data are expressed as means ± SD (three biological replicates for each condition). ^#^
*p* < 0.05 is relative to corresponding untreated HMECs and * *p* < 0.05 is relative to cells transfected with scrambled siRNA. The data underwent two-way ANOVA; (**g**) Endothelial permeability in response to AMPK activation, LPS challenge, and AMPK inhibitor SBI0206965. HMECs were grown on Transwell inserts for 72 h and treated or not treated with SBI0206965 (10 μM) for 15 min before undergoing 991 and LPS treatments. Data are expressed as means ± SD (three biological replicates for each condition). ^#^
*p* < 0.05 is relative to respective non-treated HMECs and ^$^
*p* < 0.05 is relative to LPS-only treated HMECs. The data underwent two-way ANOVA.

**Figure 4 ijms-21-05581-f004:**
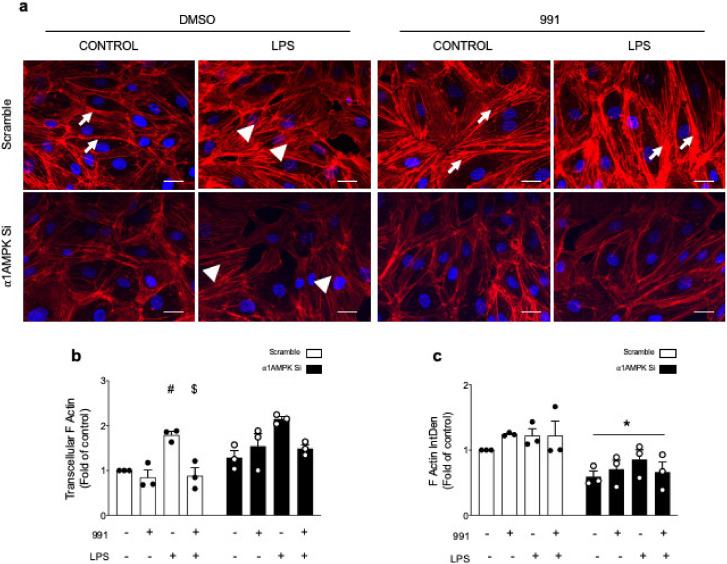
AMPK Activation Reinforces the Cortical Actin Network and Counters Stress-Fiber Formation During LPS Challenge. HMECs were transfected with control non-targeting siRNA or α1AMPK siRNA (50 nM) for 48 h and treated with DMSO or 991 (1μM) for one hour. They were then exposed to LPS (50 μM) for six hours. After fixation, HMECs were stained with Alexa Fluor 568 Phalloidin. (**a**) Representative fluorescence microscopy images of F-actin. Arrows indicate peripherical actin; arrowheads indicate centralized stress fibers; (**b**) Quantification of transcytoplasmic actin filaments; (**c**) Quantification of global F-actin amounts, which were assessed by integrated density measurement. Nuclei are stained with DAPI. Scale bar, 25 μm. Data are expressed as means ± SD (three biological replicates for each condition). ^#^
*p* < 0.05 is relative to corresponding untreated HMECs, ^$^
*p* < 0.05 is relative to corresponding LPS-treated HMECs, and * *p* < 0.05 is relative to cells transfected by scrambled siRNA. The data underwent two-way ANOVA.

**Figure 5 ijms-21-05581-f005:**
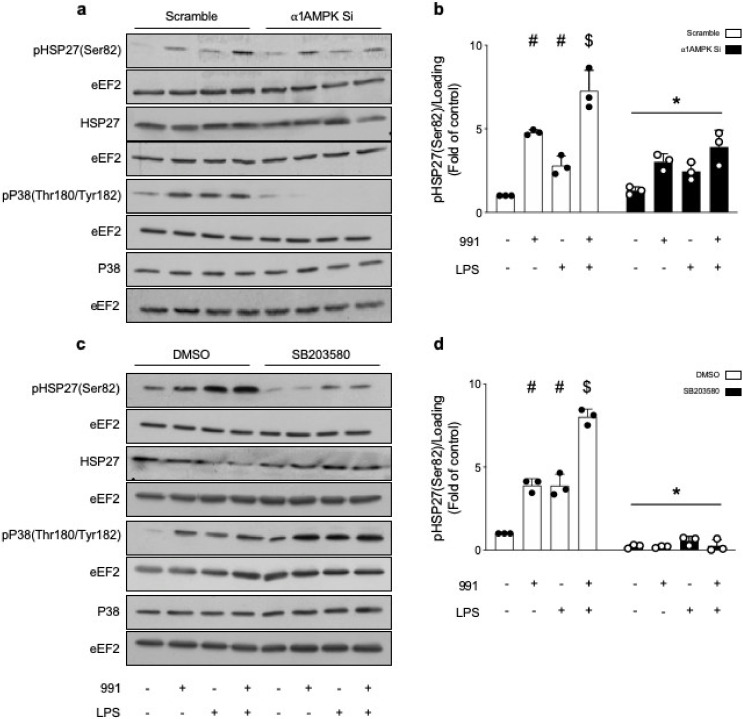
AMPK Activation Leads to Increased HSP27 Phosphorylation, via a p38 MAPK-Dependent Way. (**a**,**b**) HMECs were transfected with control non-targeting siRNA or α1AMPK siRNA (50 nM) for 48 h and then treated with DMSO or 991 (1 μM) for one hour. They were then exposed to LPS (50 μM) for six hours; (**c**,**d**) HMECs were preincubated with DMSO or with the p38 MAPK inhibitor SB203580 (10 μM) for 15 min. Then, they were treated with 991 (1 μM) or DMSO for one hour and subsequently, either exposed or not exposed to LPS (50 μM) for six hours. (**a**–**d**) Cell lysates were submitted to western blot analysis and probed with total HSP27, total p38 MAPK, phospho-HSP27 and phospho-p38 MAPK Abs. eEF2 was used as a loading control. Representative Western blots (**a**,**c**) and quantification of HSP27 (Ser82) phosphorylation (**b**,**d**) are shown. Data are expressed as means ± SD (three biological replicates for each condition). ^#^
*p* < 0.05 is relative to corresponding non-treated HMECs, ^$^
*p* < 0.05 is relative to corresponding LPS-treated HMECs, and * *p* < 0.05 is relative to cells transfected with non-targeting siRNA or that were treated with the vehicle. The data underwent two-way ANOVA.

**Figure 6 ijms-21-05581-f006:**
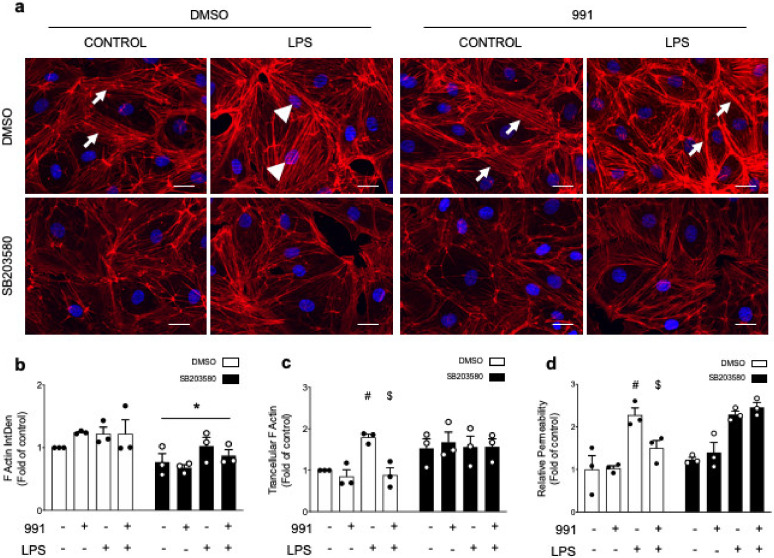
AMPK Activation Regulates Actin Organization and Cell Permeability via the p38 MAPK/HSP27 Pathway. (**a**–**c**) HMECs were fixed and stained with Alexa Fluor 568 Phalloidin. (**a**) Representative fluorescence microscopy images of F-actin. Arrows indicate peripherical actin; arrowheads indicate centralized stress fibers; (**b**) Quantification of area occupied by transcytoplasmic actin filaments; (**c**) Quantification of global F-actin amounts, which were assessed by integrated density measurement. Nuclei are stained with DAPI. Scale bar, 25 μm. Data are expressed as means ± SD (three independent experiments for each condition). * *p* < 0.05 is relative to cells treated with DMSO. The data underwent two-way ANOVA; (**d**) Endothelial permeability in response to AMPK activation, LPS challenge, and p38 MAPK inhibitor SB203580. HMECs were grown on Transwell inserts for 72 h and treated or not treated with SB203580 (10 μM) for 15 min before undergoing 991 and LPS treatments. Data are expressed as means ± SD (three biological replicates for each condition). ^#^
*p* < 0.05 is relative to respective non-treated HMECs and ^$^
*p* < 0.05 is relative to LPS-only treated HMECs. * *p* < 0.05 is relative to cells treated with the vehicle. The data underwent two-way ANOVA.

**Figure 7 ijms-21-05581-f007:**
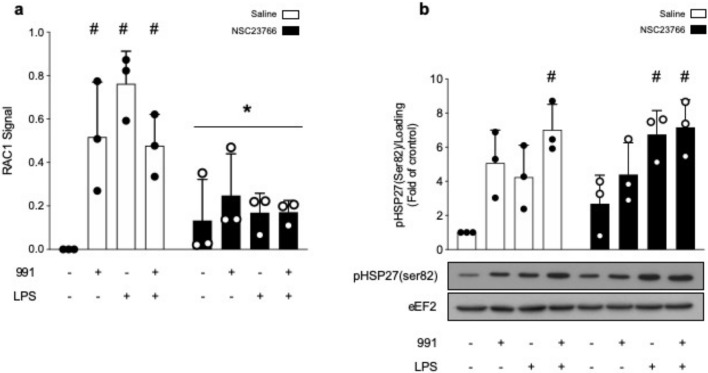
AMPK Regulates Actin Organization Independently of Rac1. (**a**,**b**) HMECs were treated with saline or the Rac1 inhibitor NSC23766 (100μM) for 15 min before adding DMSO or 991 (1μM) for one hour. Then, they were exposed to LPS (50μM) for six hours. (**a**) Lysates were engaged in G-LISA assays to precisely monitor Rac1 activity. Quantification represents the differential signal, compared to basal, in fold of control; (**b**) Representative Western blots and quantification of HSP27 (Ser82) phosphorylation. eEF2 was used as a loading control. Data are expressed as means ± SD (three biological replicates for each condition). ^#^
*p* < 0.05 is relative to corresponding non-treated HMECs. * *p* < 0.05 is relative to cells treated with the vehicle. The data underwent two-way ANOVA.

**Figure 8 ijms-21-05581-f008:**
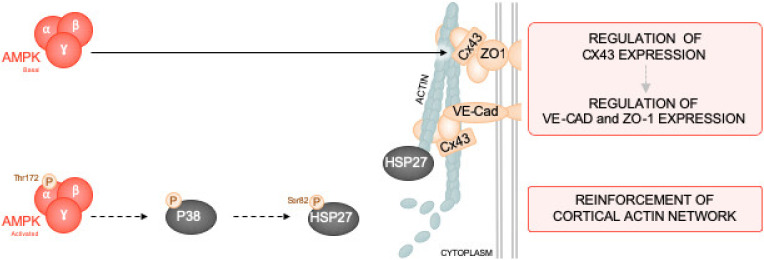
Molecular mechanisms involved in endothelial barrier protection by α1AMPK. α1AMPK is essential in maintaining the proper expression and architecture of IEJs in basal conditions. It regulates Cx43 expression, which through its scaffolding functions, may contribute to ZO-1 and VE-Cad organization at plasma membrane. On the other hand, the specific and direct pan-AMPK activator 991 protects HMECs against LPS-induced hyperpermeability. The mechanism involves activation of the p38 MAPK/HSP27 pathway and subsequent cortical actin polymerization, which reinforces IEJs anchorage at the plasma membrane and subsequent intercellular tethering. Abbreviations: AMP-activated protein kinase (AMPK), p38 mitogen-activated protein kinase (P38), Heat shock protein 27 (HSP27), VE-cadherin (VE-Cad), Zonula occludens-1 (ZO-1), Connexin 43 (Cx43).

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
