# Peer review of "α1AMP-Activated Protein Kinase Protects against Lipopolysaccharide-Induced Endothelial Barrier Disruption via Junctional Reinforcement and Activation of the p38 MAPK/HSP27 Pathway"

_ijms, 2020, doi:10.3390/ijms21155581_

Round 1

Reviewer 1 Report

Role of α1 AMPK is well defined in endothelial permeability. However, associated signaling mechanisms are unknown. In this manuscript Ange et.al., demonstrated the important role of p38MAPK/HSP27 in α1 AMPK induced endothelial barrier stabilization. Manuscript is written well. However, there are major issues with the manuscript, which needs to be addressed.   In manuscript title word "Eeinforcement" . What does it mean?   To confirm the changes in cell junction protein association, Phospho-Tyr658 VE-cadherin should be used in experiments. AMPK is known to reduce phosphorylation of VE-cadherin and thus enhances the barrier function.   Figure 2: Authors have mentioned the important role of CX43 in AMPK signaling. To confirm this, a permeability assay should be performed on CX43 deficient cells in the presence of 991.    In addition to α1 AMPK, α2 AMPK also plays a critical role in endothelial permeability as reported by others. However, authors have not talked about α2 AMPK in any place.    There are discrepancies in results, Figure 3g: treatment with 991 signaificantly reduced the basal permeability. However, in figure 6d, 991 fails to do so.   In discussion, statement: Given that p38 MAPK is a downstream effector of AMPK in endothelial cells (77). This paper (77) does not say that p38 is a downstream effector of AMPK. Here AMPK activator inhibits the TNF induced activation of p38. At basal level, there is no difference. Only in isolated rat heart papillary muscles, AMPK activator activates p38 activation (ref: 81). Author should carefully choose existing literature for citation.    Figure 5C: SB203580 fails to show any inhibition of p38 activation. Even it shows enhanced activation in all samples treated with SB203580.    Figure 5A needs levels of p38 activation in cells treated with scramble or α1AMPKsiRNA since p38/hsp27 activation is the only novel finding of the manuscript.   Figue 6D: SB203580 fails to show any protection against LPS induced hyper-permeability. Role of p38 activation in LPS induced permeability is well established in previous reports.    Significance stars are missing from figure 6C.        

Reviewer 2 Report

The manuscript by Angé et al. demonstrates that α1AMPK deficiency is associated with decreased CX43, ZO-1, and VE-Cad expression in HMECs. Compound 991-activated α1AMPK protects against LPS-induced endothelial barrier disruption via activation of the p38 MAPK and HSP27 pathway, which is independent of the small GTPase Rac1. Specific comments are as follows:

  1. In the introduction, to increase novelties of the manuscript, please add some sentences to indicate what was unknown for AMPK to regulate Cx43 since AMPK was shown to suppress Cx43 in other cell types. 

  1. In the results, “α1. AMPK” should be changed to “α1AMPK” in the subtitles.

  1. In Figure 1, please show immunoblots for the phosphorylation of AMPK at Thr172 after 991 and LPS treatments.

  1. In Figure 1 and Figure 3, the data show that that VE-Cad, ZO-1, and Cx43 protein expression are significantly downregulated in α1AMPK-depleted cells, although they are not affected by 991 or LPS treatment (Figure 1b and 1c), and membrane staining quantification for VE-Cad, ZO-1, and Cx43 is significantly lower in LPS-treated cells, compared to control cells (Figure 3b, 3d, 3f). Please explain why 991 would not affect VE-Cad, ZO-1, and Cx43 expression, but α1AMPK-depletion significantly downregulated VE-Cad, ZO-1, and Cx43 expression. Please also explain why membrane staining quantification for VE-Cad, ZO-1, and Cx43 was significantly lower in LPS-treated cells, but VE-Cad, ZO-1, and Cx43 protein expression were not affected by LPS treatment.

  1. In Figure 5 and Figure 6, how to explain why LPS can enhance HSP27 phosphorylation? It is confusing that AMPK-activated HSP27 (AMPK-increased HSP27 phosphorylation) protects against LPS-induced endothelial barrier impairment but LPS itself can also enhance HSP27 phosphorylation. There is also a missing link between HSP27 and Cx43. Does HSP27 phosphorylation affect Cx43 expression? Please discuss or perform experiments to show how LPS or AMPK-activated HSP27 affect Cx43 expression.

  1. To elucidate more clearly about the working model of this paper, please add some descriptions and definitions (i.e. TAB) in the figure legends of Figure 8. Please also explain the question mark on the β subunit of AMPK in Figure 8.

Round 2

Reviewer 1 Report

Authors have answered all the questions.

Minor comment:

Figure 5A; Please keep total HSP27 levels in the figure as shown in manuscript ver 01. Also in figure 5B; show the total levels of HSP27 and p38 . 

Reviewer 2 Report

Authors have answered most of the questions.

Minor comments:

  1. Please add the information of pAMPK (Thr172) antibody in “1. Reagents and Antibodies“ and “4.4. Western Blotting”.
  2. The authors show that basal α1AMPK participates in the regulation of CX43, VE-Cad and ZO-1 expression, independently of its activation state. However, I would expect that it should be “activated α1AMPK” participates in the regulation of CX43, VE-Cad and ZO-1 expression. Transfection of α1AMPK-targeting siRNA abrogates α1AMPK expression as well as eliminates activated α1AMPK, therefore resulting in reduced CX43, VE-Cad and ZO-1 expression. Since 911 can activate α1AMPK, I would expect that CX43, VE-Cad and ZO-1 expression would be induced after 911 treatment. Please explain and discuss why α1AMPK can regulate CX43, VE-Cad and ZO-1 expression, independently of its activation state in the result or discussion section.
  3. Please add the description of LPS effects on regulating the relative amount of VE-Cad, ZO-1 and Cx43 localized at plasma membrane due to their mechanical disruption and likely internalization, but not due to the decreased expression levels of VE-Cad, ZO-1 and Cx43 in the result section.
  4. Please add the description of the controversial role of the p38 MAPK/HSP27 pathway in the regulation of endothelial permeability in the result or discussion section.
  5. Please add the description of the potential link between HSP27 and Cx43 in the discussion section.
